# AutoLink: Self-supervised Learning of Human Skeletons and Object Outlines by Linking Keypoints

**Xingzhe He**      **Bastian Wandt**      **Helge Rhodin**
University of British Columbia
{xingzhe, wandt, rhodin}@cs.ubc.ca

## Abstract

Structured representations such as keypoints are widely used in pose transfer, conditional image generation, animation, and 3D reconstruction. However, their supervised learning requires expensive annotation for each target domain. We propose a self-supervised method that learns to disentangle object structure from the appearance with a graph of 2D keypoints linked by straight edges. Both the keypoint location and their pairwise edge weights are learned, given only a collection of images depicting the same object class. The resulting graph is interpretable, for example, AutoLink recovers the human skeleton topology when applied to images showing people. Our key ingredients are i) an encoder that predicts keypoint locations in an input image, ii) a shared graph as a latent variable that links the same pairs of keypoints in every image, iii) an intermediate edge map that combines the latent graph edge weights and keypoint locations in a soft, differentiable manner, and iv) an inpainting objective on randomly masked images. Although simpler, AutoLink outperforms existing self-supervised methods on the established keypoint and pose estimation benchmarks and paves the way for structure-conditioned generative models on more diverse datasets. Project website: https://xingzhehe.github.io/autolink/.

## 1   Introduction

Object structure representations are widely used in modern computer graphics and computer vision techniques, including keypoints for image generation [67, 68, 94] and skeletons for 3D reconstruction [26, 41, 123, 83, 99]. However, the structure is usually supervised on large annotated datasets [61, 2, 3, 71] or via hand-crafted parametric models [65, 84, 111, 80, 5, 59]. Neither approach generalizes well to new domains and both require additional manual annotation whenever more detail is needed [88].

Our goal is to reconstruct the keypoint locations of an object by learning from an unlabelled image collection, thereby sidestepping the generalization problem. Our key idea is to leverage that the same object shares the same topology by introducing an explicit graph that links the same pairs of keypoints in all instances. By contrast, existing self-supervised keypoint learning methods model objects as a set of independent parts. Their consistency over different instances of the same object is encouraged by either enforcing parts to follow hand-crafted image transformations [102, 126, 42, 66, 37, 62] or by adding implicit bias in the network architecture that encodes such spatial equivariances [35, 34]. Only [43, 93] use an explicit skeleton representation, but both require predefined topology, rely on video input, and [43] is trained in a CycleGAN setting that still requires manually labeled examples.

We propose a simple yet effective method to learn both the keypoints and their links without supervision in terms of a sparse graph serving two purposes. First, the graph acts as a bottleneck that can only store structural information disentangled from appearance. Second, it forms a constraint that associates observations across training images. We enforce the same topology across instances of the same class by learning a shared graph with a single set of edge weights. In the absence of labels,

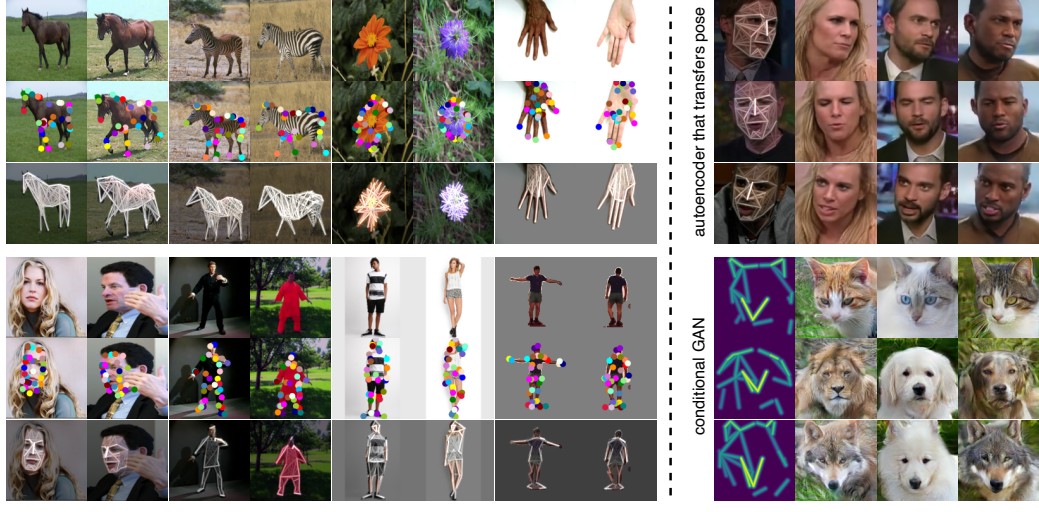

autoencoder that transfers pose

conditional GAN

(a) detected keypoints and visualized graph representation                    (b) applications

Figure 1: **Teaser.** (a) AutoLink applies to diverse collections of images and automatically yields keypoints linked to a graph without ground truth. It recovers animal and human poses and object shapes in settings where previous methods struggle, including cluttered backgrounds, structured stripe textures, articulated fingers, and detailed faces. (b) Example applications are conditional image generation with autoencoders (top) and GANs (bottom) that are driven by the learned keypoints.

we train AutoLink with only an autoencoder reconstruction objective. Since the graph bottleneck should not model appearance, we additionally feed the decoder with the input image after masking the majority of its pixels. In turn, it is important for the disentanglement that the heavily masked image contains appearance without leaking structural information. This is the case as inpainting methods are unable to infer the original image precisely from only sparse pixel colors [129, 124] unless conditioning on structure representations, such as edge maps [110, 75, 58, 56, 31]. Therefore, by forcing the autoencoder to reconstruct the original image, the detector converges to generate representative structures of images.

We demonstrate on 4 benchmarks that the trained detector has a significantly improved keypoint localization accuracy and on 6 additional datasets that it applies to a broader set of images spanning portraits, persons, animals, hands, and flowers, which we attribute to the explicit modeling of links in the graph. Figure 1 shows the diverse set of image domains it applies to, including challenging textures and uncontrolled background, how both skeleton representations as well as object outlines are learned by varying the number of keypoints, and exemplifies applications to controlled image generation.

**Ethics - Risks.** The estimated keypoints and edges could be abused for deep fakes as the driving signal for generative models or for unwanted surveillance applications. However, our method works towards improved generality, including objects and animals, and does not improve upon supervised models that already exist in high detail for humans. **Benefits.** Since our method is entirely self-supervised, it can be applied to a diverse set of persons, objects, animals, or situations that have not yet been labeled.

## 2    Related Work

Most representation learning methods focus on generic feature vectors for entire images to initialize deep networks for improved object classification [108, 11, 79, 32, 33]. By contrast, our method introduces explicit object structure. We review the most related approaches in the following.

**Self-supervised Keypoints Detection.** The most common idea to discover keypoints in an unsupervised manner is to rely on the notion that keypoints move as the image changes. Various constraints have been used to enforce that the keypoints follow a known transformation, including view changes in a multi-view recording [101, 86, 87] or the natural motion in videos [23, 95, 55, 74, 20, 53, 43].

When only single images are available, artificial image deformation is applied, either from randomized [102, 126, 42, 66] or learned [107, 116] transformations within a pre-defined deformation space. However, learned keypoints may model the background [126, 95] and struggle with large pose variation [37] as image deformations do not separate foreground from background, which are usually tuned for each dataset and bound to be small. Most models leverage multi-branch network architectures to encode the structure and appearance separately and utilize multiple losses that need to be balanced. By contrast, we do not apply artificial transformations, use a single branch, and a single loss which eases and stabilizes training. To overcome the need for artificial image deformation, He et al. [34, 35] exploit GANs to generate images along with corresponding keypoints and later use them to train a detector. However, this leads to even more complex network architectures and comes with instabilities in GAN training, limiting their applicability to complex objects like human bodies.

**Skeleton Representations.** Bone maps representing the keypoints connectivity as affinity fields [9] or via explicit offsets [82] are used in supervised human, animal, and object pose estimation. We use a similar edge map representation but learn both the location and linking from scratch without annotations. In the weakly-supervised setting, Jakab et al. [43] exploit CycleGAN [130] to translate between image and edge maps. The graph connectivity is predefined to the human skeleton and edges are supervised by a large dataset of unpaired ground truth object edges, which can come from a different dataset but are manually annotated. Schmidtke et al. [93] overcome the manual labeling by deforming a template skeleton. However, they both require the known connectivity of the keypoints and videos for training while ours learns both the keypoints and connectivity from a collection of single images. Noguchi et al. [78] generate a skeleton heuristically by linking the centers of part-wise learned Signed Distance Fields [70]. However, they require videos without the background of the same object, and the learned skeleton does not generalize to other objects of the same class.

**Object Sketch Learning.** Sketches are made of strokes drawn by a pen. It is a concise and abstract representation, which can be used in object recognition [121, 115] and image retrieval [104, 122, 114]. There are two common sketch representations used in neural models [113]: black-white raster images [105, 91, 120], often used for image-to-image translation [40, 130, 81, 48] and sequences of points (pen coordinates) [30, 24, 90], which is usually used by recurrent generation models [30, 12, 8, 47, 19, 28]. This graph representation is similar to ours. However, instead of learning to mimic human drawings, ours directly predicts both the keypoints and their connectivity on real natural images.

**Structure-enhanced Image Inpainting.** When key parts are missing in an image, e.g., eyes on faces or arms of humans, it is hard for inpainting networks to imagine the content accurately from scratch. Therefore, additional structural cues are detected to guide the subsequent image generation. The cues can be supervised segmentation masks [128, 60, 31, 98], foreground contours [110], and landmarks [56, 125, 117], or automatically extracted edges [75, 58, 45, 112, 7] and low-frequency image components [106, 85]. Our reconstruction objective can be seen as such two-stage inpainting, but self-supervised and with the image edges replaced with the learned graph edge representation.

**Self-supervised Foreground Segmentation.** Traditional methods use color [131], contrast [13], and hand-crafted features [44] to cluster foreground pixels. A recent trend is exploiting inpainting techniques to segment the foreground. Chen et al. [10] and Arandjelović et al. [4] use a GAN to inpaint the background at the predicted segmentation mask, assuming that the object texture can be changed without changing the data distribution due to the independence of foreground and background. Yang et al. [118] propose Contextual Information Separation (CIS), a general objective to segment the foreground by maximizing the error of inpainting the mask and its complement. It was first applied to optical flow maps and subsequently to RGB images by [92, 119, 51]. When the object is small compared to the background, an additional object detection module [51, 18] or multi-view [52] information is required. Different from these previous methods, we utilize a form of inpainting to learn sparse keypoints instead of segmentation. Our learned edges form a sparse foreground shape, but further extensions would be necessary to transfer from the edge maps to the boundary-aligned foreground segmentation.

## 3 Method

We leverage that the objects in the dataset share the same topology and can be represented as a graph that connects keypoints by a shared set of edges. To learn the keypoints and edges, we design an

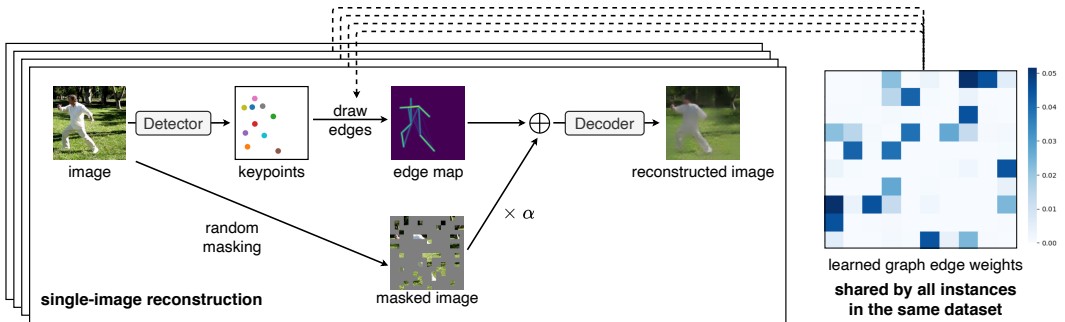

Figure 2: **Overview.** Given an image, we detect keypoints and draw differentiable edges between keypoints according to the learned graph edge weights that is visualized as a color matrix. The method is self-supervised in that the latent edge map and keypoints are learned by reconstructing the masked input images. Note that keypoints are image specific and edge maps are shared.

autoencoder that aims to accurately reconstruct the input image, with the graph as the intermediate representation. To encode the input image into a graph, we detect the keypoints and create an edge heatmap based on learnable edge weights. To mostly obtain appearance information, we mask out the majority of the image, which randomizes the structure information and reduces the remainder to a very low level. The edge heatmap is combined with the masked image to reconstruct the original image. Since the missing structure is important to reconstructing the original image, the network is forced to learn the structure of the object in a self-supervised manner. Figure 2 shows an overview of our method.

Formally, given an image $\mathbf{I} \in \mathbb{R}^{H \times W \times 3}$ with height $H$ and width $W$ we aim to learn a set of keypoints $\{\mathbf{k}_i\}_{i=1}^{K}$, where $\mathbf{k}_i \in [-1, 1] \times [-1, 1] \subset \mathbb{R}^2$ is the normalized keypoint coordinate, and $K$ is the number of keypoints. We use a ResNet with upsampling [109] to detect keypoints. Afterward, we draw a differentiable edge [73] between each pair of keypoints (details below). The edge map $\mathbf{S} \in \mathbb{R}^{H \times W}$ is concatenated along the channel dimension with the randomly masked image $\mathbf{I}_m \in \mathbb{R}^{H \times W \times 3}$ and fed into a UNet [89] to obtain the reconstructed image $\mathbf{I}'$. The detailed network architectures can be found in Appendix D.

### 3.1 Image Structure Representation

In this section, we introduce the generation of keypoints and the edge map from the image. Let $\mathbf{H} \in \mathbb{R}^{H \times W \times K}$ be the $K$ heatmaps generated by a ResNet with upsampling [109] from the image $\mathbf{I}$. The keypoint $\mathbf{k}_i$ is calculated by the differentiable soft-argmax function,

$$\mathbf{k}_i = \sum_{\mathbf{p}} \frac{\exp(\mathbf{H}(\mathbf{p}))}{\sum_{\mathbf{p}}(\exp \mathbf{H}(\mathbf{p}))} \mathbf{p}, \tag{1}$$

where $\mathbf{p} \in [-1, 1] \times [-1, 1]$ is the normalized pixel coordinates.

Given two keypoints $\mathbf{k}_i, \mathbf{k}_j$, we draw a differentiable edge map $\mathbf{S}_{ij}$, where values are 1 on the edge linked by the two keypoints and decrease exponentially based on the distance to the line. Formally, the edge map $\mathbf{S}_{ij}$ is a Gaussian extended along the line [73], defined as

$$\mathbf{S}_{ij}(\mathbf{p}) = \exp\left(d_{ij}^2(\mathbf{p})/\sigma^2\right), \tag{2}$$

where $\sigma$ is a hyperparameter controlling the thickness of the edge, and $d_{ij}(\mathbf{p})$ is the $L_2$ distance between the pixel $\mathbf{p}$ and the edge drawn by keypoints $\mathbf{k}_i$ and $\mathbf{k}_j$,

$$\mathbf{d}_{ij}(\mathbf{p}) = \begin{cases} \|\mathbf{p} - \mathbf{k}_i\|_2 & \text{if } t \leq 0, \\ \|\mathbf{p} - (\mathbf{k}_i + t\mathbf{k}_j)\|_2 & \text{if } 0 < t < 1, \\ \|\mathbf{p} - \mathbf{k}_j\|_2 & \text{if } t \geq 1, \end{cases} \quad \text{where} \quad t = \frac{(\mathbf{p} - \mathbf{k}_i) \cdot (\mathbf{k}_j - \mathbf{k}_i)}{\|\mathbf{k}_i - \mathbf{k}_j\|_2^2}. \tag{3}$$

We assign a weight $w_{ij} > 0$ to each edge, which is enforced to be positive by SoftPlus [22]. This weight is learned during training and shared across all object instances in a dataset. Finally, we take

the maximum at each pixel of the heatmaps to obtain the final edge map $\mathbf{S} \in \mathbb{R}^{H \times W}$,

$$\mathbf{S}(\mathbf{p}) = \max_{ij} w_{ij} \mathbf{S}_{ij}(\mathbf{p}). \tag{4}$$

Taking the maximum at each pixel avoids the entanglement of the edge weights and the convolution kernel weights, which is further explained in Section 4.4.

## 3.2 Image Reconstruction

The masked image $\mathbf{I}_m$ is generated by first uniformly dividing the image $\mathbf{I}$ into a $16 \times 16$ grid, and randomly masking out 80% of the grid cells, similar to [33]. We concatenate the masked image with the edge map and feed it into a UNet decoder [89] to reconstruct the original image,

$$\mathbf{I}' = \text{Decoder}(\alpha \mathbf{I}_m \oplus \mathbf{S}) \tag{5}$$

where $\oplus$ means concatenation along the channel dimension and $\alpha$ is a learnable parameter that compensates for the change of the edge weight magnitude during training. $\alpha$ is initialized to 1. We found this parameter to be helpful in training stability. Different to [33], we condition on an edge map. Different to [43], we have no ground truth for the edge map. Our edge map is an unobserved latent variable. Thus we only minimize the difference of the original image and the reconstructed image by the perceptual loss [46],

$$\mathcal{L} = \frac{1}{N} \sum_{i=1}^{N} \|\Gamma(I_i) - \Gamma(I_i')\|_2^2 \tag{6}$$

where $N$ is the number of examples and $\Gamma$ is the feature extractor. The perceptual loss is believed to measure the structure similarity [46, 27, 21], and leads to more robust training [42, 43].

## 3.3 Implementation Details

We use the Adam optimizer [54] with a learning rate of $10^{-4}$ with $\beta_1 = 0.9$, $\beta_2 = 0.99$. The batch size is 64. We train for 20k iterations. It takes 3 hours to train on a single V100 GPU. All images are resized to $128 \times 128$. The learning rate for the edge weights is multiplied by 512 due to the small gradient of SoftPlus [22] when the value is close to 0. To show the robustness of our model, we report all experiments on the sampling strategy of masking 80% of the $16 \times 16$ patches. We perform experiments with the same edge thickness of $\sigma^2 = 5e - 5$ for all benchmark datasets. We train 10 times and report the mean and the standard deviation of the evaluation metrics. Although it already outperforms other work in most experiments, we also tune thicknesses to each individual dataset, as others did for their hyperparameters, which further improves the results. The tuned thicknesses can be found in Appendix B. The only other hyperparameter is the number of keypoints, which we set to that of the established benchmarks for quantitative comparisons, ranging from 4 to 32 points.

## 4 Experiments

In this section, we compare our results to the related methods, showing that our model is simple yet effective. Besides, we perform a number of ablation studies on hyperparameters and algorithm variants, exhibiting the robustness of our model and justifying the necessity of every model component.

### 4.1 Datasets and Evaluation Metrics

**CelebA-aligned** [63] contains 200k celebrity faces aligned in center. We follow [102] splitting it into three subsets: CelebA training set without MAFL (160k images), MAFL training set (19k), MAFL test set (1k). We train our network on the CelebA training set without MAFL. To quantitatively evaluate the consistency of our predicted keypoints, we follow [102] training a linear regression without bias from our detected keypoints to the ground truth keypoints on the MAFL training set and reporting the mean $L_2$ error normalized by inter-ocular distance on the MAFL test set.

**CelebA-in-the-wild** [63] contains celebrity faces in unconstrained conditions. We follow [37] and first split it into three subsets as for CelebA-aligned, and then remove the images where a face covers less than 30% of the area, which results in 45,609 images for model training, 5,379 with keypoint labels for regression, and 283 for testing. The evaluation metric is the same as CelebA-aligned.

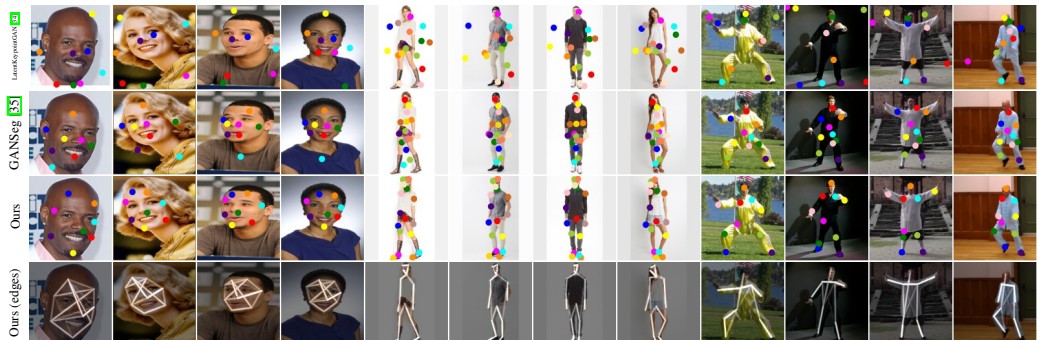

Figure 3: **Qualitative comparison on detected keypoints**. Our model is more robust on wild face poses, and depicts more details on human bodies compared to [34] and [35]. For example, the feet poses in the middle four images are clearly detected.

Table 1: **Landmark detection on CelebA**. The metric is the landmark regression (without bias) error in terms of $L_2$ distance normalized by inter-ocular distance (lower is better). While all methods perform well on aligned CelebA, ours is more robust on Wild CelebA. The sign $\star$ means being reported by [37] and † means being reported by [62].

| Method | Type | Aligned (K=10) | Wild (K=4) | Wild (K=8) |
|---|---|---|---|---|
| DFF [17] by [37] | Part Segmentation | - | - | 31.30% $\star$ |
| SCOPS [37] (w/o saliency) | Part Segmentation | - | 46.62% | 22.11% |
| SCOPS [37] (w/ saliency) | Part Segmentation | - | 21.76% | 15.01% |
| Liu et al. [62] | Part Segmentation | - | 15.39% | 12.26% |
| Huang et al. [36] (w/ detailed label) | Part Segmentation | - | - | 8.40% |
| GANSeg [35] | Part Segmentation | 3.98% | **12.26%** | **6.18%** |
| Thewlis et al. [102] | Landmark | 7.95% | - | 31.30% $\star$ |
| Zhang et al. [126] | Landmark | 3.46% | - | 40.82% $\star$ |
| LatentKeypointGAN [34] | Landmark | 5.85% | 25.81% | 21.90% |
| Lorenz et al. [66] | Landmark | 3.24% | 15.49% † | 11.41% † |
| IMM [42] | Landmark | **3.19%** | 19.42% † | 8.74% † |
| LatentKeypointGAN-tuned [34] | Landmark | 3.31% | 12.10% | 5.63% |
| Ours (general) | Landmark | 3.92±0.69% | 7.72±0.47% | 5.66±0.29% |
| Ours (thickness-tuned) | Landmark | 3.54% | **6.11%** | **5.24%** |

**Human3.6m** [39] contains human activity videos in static backgrounds. We follow [126] considering six activities (direction, discussion, posing, waiting, greeting, walking), and using subjects 1, 5, 6, 7, 8, 9 for training and 11 for testing. This results in 796,648 images for training and 87,975 images for testing. The evaluation metric is the regressed (without bias) mean $L_2$ error normalized by the image size. We remove the background as in [126, 66] to make a fair comparison to others. To underline the robustness against structured backgrounds we also report the numbers including background.

**DeepFashion** [64] contains 53k in-shop clothes images. We follow [66] only keeping the full-body images. We use 10604 images for training and 1179 images for testing as in [6]. We use the keypoints generated by AlphaPose [25] as the ground truth. The evaluation metric is Percentage of Correct Keypoints of d=6 pixels in resolution $256 \times 256$.

**Taichi** [96] contains 3049 training videos and 285 test videos of people performing Tai-Chi, with the various appearance of foreground and background. We follow [97] using 5000 and 300 images (not contained in training data) for training a linear regression and for testing, respectively. The evaluation metric, mean average error (MAE), is calculated as the sum of the $L_2$ error in resolution $256 \times 256$.

**CUB-200-2011** [103] consists of 11,788 images of birds. We follow two established protocols [66, 15] to evaluate our method: 1) Images are cropped based on the bird landmarks, aligned to face to the left [66], and seabirds are removed; 2) Birds are cropped based on the given bounding box and the train/val/test split of [15] is used. In both cases, the evaluation metric is the regressed (without bias) mean $L_2$ error normalized by the cropped image size.

**Flower** [77], **11k Hands** [1], **Horses** [130], and **Zebras** [130] are used for qualitative experiments. **VoxCeleb2** [16] and **AFHQ** [14] are used for pose transfer and conditional image generation,

Table 2: **Landmark detection on Human Body**. Our model outperforms all the other unsupervised baselines. The metric for each dataset follows the corresponding description in the text. The sign † means being reported by [97] and the sign ⋆ means being reported by [6]. The number of keypoints is $K = 16$ for Human3.6m and DeepFashion and $K = 10$ for Taichi.

| Method | Supervision | Human3.6m ↓ | DeepFashion ↑ | Taichi ↓ |
|---|---|---|---|---|
| Jakab et al. [43] | video & unpaired ground truth | 2.73 | - | - |
| Newell et al. [43] | paired ground truth | **2.16** | - | - |
| DFF [17] | testing dataset | - | - | 494.48 † |
| SCOPS [37] | saliency maps | - | - | 411.38 † |
| Siarohin et al. [97] | videos | - | - | 389.78 |
| Zhang et al. [127] | videos | - | - | **343.67** |
| Zhang et al. [126] | videos | 4.14 | - | - |
| Schmidtke et al. [93] | video & T-pose template | 3.31 | - | - |
| Sun et al. [100] | videos | **2.53**±0.06 | - | - |
| Thewlis et al. [102] | unsupervised | 7.51 | - | - |
| Zhang et al. [126] | unsupervised | 4.91 | - | - |
| LatentKeypointGAN [34] | unsupervised | - | 49% | 437.69 |
| Lorenz et al. [66] | unsupervised | 2.79 | 57% ⋆ | - |
| GANSeg [35] | unsupervised | - | 59% | 417.17 |
| Ours (general) | unsupervised | 2.81±0.07 | 65±1.2% | 337.50±25.08 |
| Ours (thickness-tuned) | unsupervised | **2.76** | **66%** | **316.10** |

Table 3: **Landmark detection on CUB Birds**. Our model outperforms most other baselines and achieves comparable results with the ones using ground truth segmentation masks. The metric is the landmark regression (without bias) error of $L_2$ distance normalized by the image size (lower is better). A star ⋆ means being reported by [15], † means being reported by [37], and ‡ means tested by us with their official code; all other numbers are taken from the respective papers. The number of keypoints is $K = 10$ for CUB-aligned and $K = 4$ for CUB-001, CUB-002, CUB-003, and CUB-all.

| Method | Supervision | CUB-aligned ↓ | CUB-001 ↓ | CUB-002 ↓ | CUB-003 ↓ | CUB-all ↓ |
|---|---|---|---|---|---|---|
| SCOPS [37] | GT silhouette | - | 18.3 ⋆ | 17.7 ⋆ | 17.0 ⋆ | 12.6 ⋆ |
| Choudhury et al. [15] | GT silhouette | - | **11.3** | **15.0** | **10.6** | **9.2** |
| DFF [17] | testing dataset | - | 22.4† | 21.6† | 22.0† | - |
| SCOPS [37] | saliency maps | - | **18.5** | **18.8** | **21.1** | - |
| Lorenz et al. [66] | unsupervised | 3.91 | - | - | - | - |
| ULD [126, 102] | unsupervised | - | 30.1† | 29.4† | 28.2† | - |
| Zhang et al. [126] | unsupervised | 5.36 | 26.9‡ | 27.6‡ | 27.1‡ | 22.4‡ |
| LatentKeypointGAN [34] | unsupervised | 5.21‡ | 22.6‡ | 29.1‡ | 21.2‡ | 14.7‡ |
| GANSeg [35] | unsupervised | **3.23** | 22.1‡ | 22.3‡ | 21.5‡ | 12.1‡ |
| Ours (general) | unsupervised | 4.15 ± 0.24 | 20.6 ± 0.54 | 20.3 ± 0.96 | 19.7 ± 0.91 | 11.6 ± 0.33 |
| Ours (thickness-tuned) | unsupervised | 3.51 | **20.2** | **19.2** | **18.5** | **11.3** |

respectively. Horses and Zebras are extracted from the CycleGAN dataset [130] by removing the images with multiple horses and aligning them to face left. Note that the horses and zebra are trained separately, yet the model learns similar structures. The train/test split of Flower follows [10]. All the other datasets follow the train/test split specified by the dataset.

## 4.2  Qualitative Analysis

We qualitatively compare our detected keypoints with other methods and show the examples of the learned edges in Figure 3. For visualization purposes, we scale the edge weights to obtain visible edges. We use the same number of keypoints as the previous method [35] for a fair comparison, which are 8 for CelebA-in-the-Wild, 16 for DeepFashion, and 10 for Taichi. We will discuss more on the choice of the number of keypoints in Ablation Study 4.4. As shown in Figure 3, our model not only detects consistent keypoints but also learns reasonable edges, such as human skeletons in DeepFashion and Taichi. For example, the feet are clearly connected with the corresponding knees, and there is no edge between the left and right hands. We show 105 images with detected keypoints and visualized graph structure for each dataset in Figure 9-23 in the Appendix, demonstrating that our model works on various classes of objects of diverse appearance and complex backgrounds.

## 4.3  Quantitative Analysis

We compare the keypoint detection results with other methods in Table 1 (CelebA), Table 2 (Human3.6, DeepFashion, Taichi), and Table 3 (CUB). Our simple model outperforms all other unsuper-

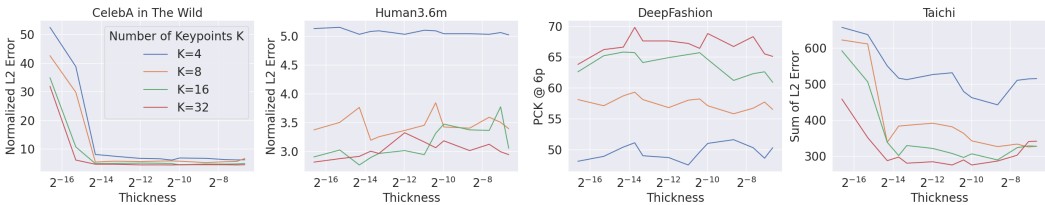

Figure 4: **Ablation tests on the number of keypoints and edge thickness.** While the model shows better performance with more keypoints, it is robust to the edge thickness.

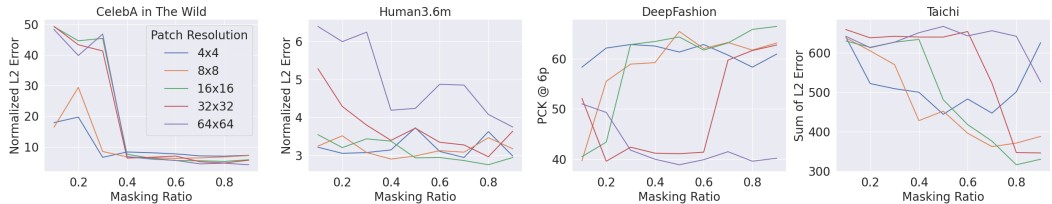

Figure 5: **Ablation tests on masking strategy.** Overall, the performance increases as the mask ratio increases. Too small or too large patch sizes can decrease the performance. Empirically, masking 80% of the $16 \times 16$ patches is a golden rule.

vised methods in all benchmarks, except [37] on three CUB subsets, which however requires saliency maps, and for the most constrained setting CelebA-aligned and CUB-aligned where all methods perform well and the results are comparable. The results on CelebA in Table 1 confirms that our model is more robust to poses in the wild. Since self-supervised part segmentation methods are usually more robust on wild faces [62, 37, 35], we also include them for comparison, demonstrating the robustness of our model over existing baselines. The results on Human3.6m and DeepFashion in Table 2 show the capability of our model to detect keypoints on human bodies of either similar or diverse appearances. The result on Taichi in Table 2 demonstrates the general applicability of our model to human bodies of diverse poses in complex backgrounds.

## 4.4 Ablation Tests

In this section, we analyze the hyperparameters and demonstrate the robustness of our model. We also discuss the possible variants of our model and show the superiority of our design.

**Number of Keypoints & Thickness**. We show ablation test results on the different numbers of keypoints and edge thicknesses in Figure 4. The exact numbers can be found in Table 5 in Appendix B. Our model shows very strong robustness to edge thickness. The accuracy remains state-of-the-art while the thickness of the edges varies by multiple orders of magnitude. On the other hand, with the increasing number of keypoints the accuracy increases. This is expected since more keypoints are able to capture structure in more detail, as shown in Figure 6. Yet, some other methods fail for a large number of keypoints [35].

**Masking Strategy**. In our standard setting, the image is divided into $16 \times 16$ patches and 80% of the patches are randomly masked. We investigate how the patch size and masking ratio affect the model performance. Figure 5 shows that a too low masking ratio enables the network to directly infer the structure from the masked image which is undesired in our case. In these cases, the network would not choose to infer a set of compact keypoints from the original image. Figure 5 illustrates that the patch size cannot be too small ($4 \times 4$) or too large ($64 \times 64$). Although in some cases, such as CelebA-in-the-Wild, a different masking strategy gives better results (4.14 vs 5.24), we choose to report a unified strategy that we mask 80% of $16 \times 16$ patches for simplicity and conciseness.

**Variants of Edge Heatmap Generation**. Besides the heatmap generation described in Method 3, we test four more ways to generate the edge maps: 1) we define the thickness as a globally learnable parameter; 2) we learn each edge thickness independently as a parameter; 3) we treat each edge heatmap as an independent channel of the feature map, instead of making them a single channel

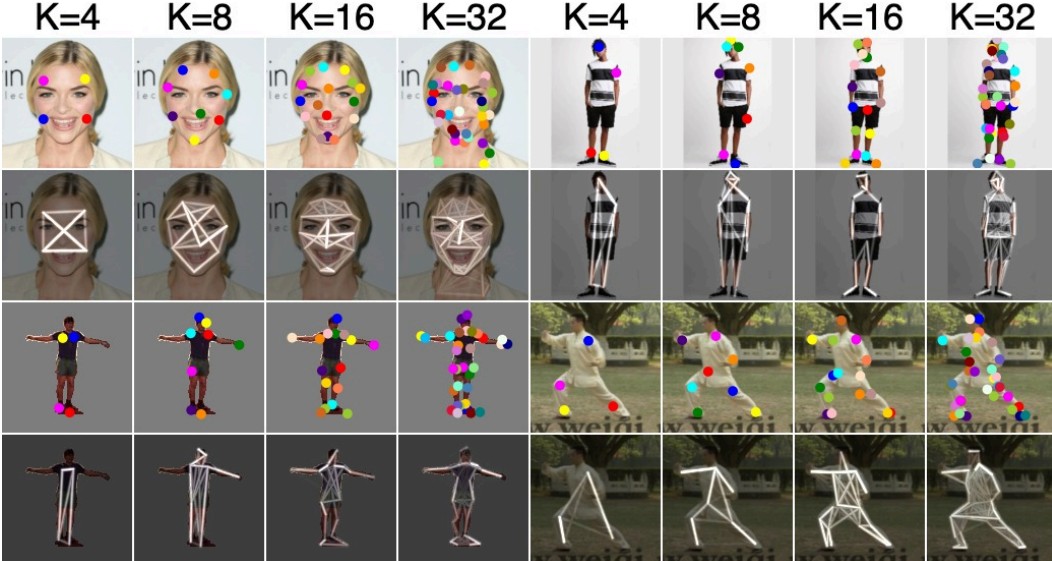

Figure 6: **Examples of different numbers of detected keypoints.** With very few keypoints, the model only models a very basic shape, such as the box on the face, and sometimes cannot fully capture the structure. For example, with K=4, only legs are modeled on humans. With abundant keypoints, it is able to model details.

Table 4: **Ablation tests on variants of edge heatmap generation**. The original design is proved to be the most robust one. Although in some cases it is not optimal, the difference is almost trivial.

| Model | CelebA in The Wild ↓ | Human3.6m ↓ | DeepFashion ↑ | Taichi ↓ |
|---|---|---|---|---|
| original model | **5.24**% | **2.76**% | 65.8% | 316 |
| fixed $\alpha$ | 6.39% | 2.87% | **66.0**% | 374 |
| shared learnable thickness | 6.12% | 3.25% | 49.1% | 425 |
| independent learnable thickness | 5.94% | 3.73% | 50.2% | **311** |
| edge-specific heatmap | 5.65% | 3.83% | 65.1% | 407 |
| only using keypoints without edges | 6.55% | 3.58% | 52.9% | 722 |

heatmap with Equation 4; 4) instead of using the edge heatmap, we generate Gaussian heatmaps for the keypoints and use Equation 4 to combine them in a single channel heatmap. The results are listed in Table 4. Overall, these variants have worse performance. Although in some cases, the model has slightly better results on specific datasets, the performance boost does not hold in general. We observe that with only keypoints without edges, the model may degenerate, as shown in Figure 7a. Interestingly, assigning each edge a different channel performs worse than simply combining all edges into a single channel. We believe it is caused by entangling the edge weights with the convolution kernel weights. As visualized in Figure 7b, there exist dummy edges that do not model the object structure. In addition, we tried to remove the learnable $\alpha$ in Equation 5, fixing $\alpha = 1$, but the overall performance decreases as shown in Table 4.

**Does Texture Matter?** We trained two networks on horses and zebras separately. As shown in Figure 1a, the horse and the zebra share similar shape structures but only one is textured. The striped texture not having a significant impact on the learned structure shows that our model primarily learns the structure instead of texture features.

**What if the model is trained on images with a structured background?** We tested on Human3.6m with a background, where all images are taken in a single room. The error is 5.02. As shown in Figure 7c, our model captures the entrance in the background. It is expected since we assume the foreground object is structured. If we apply spectral clustering [76] on the learned graph, the keypoints are clearly divided into two clusters, one for the room and one for the person.

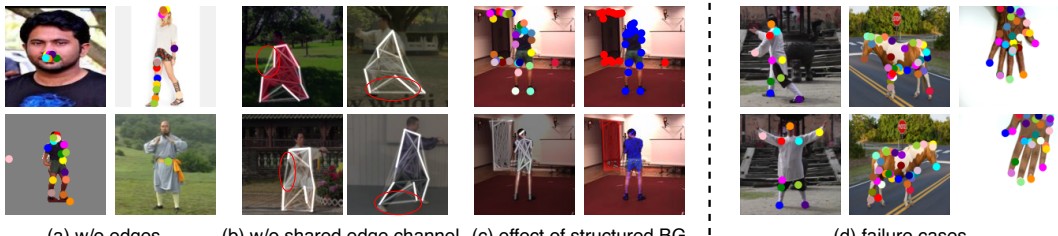

(a) w/o edges    (b) w/o shared edge channel    (c) effect of structured BG    (d) failure cases

Figure 7: (a) If we do not model the edges, the model may degenerate. (b) If we give different edges a different channel in the feature map, the model would give dummy edges. (c) If the model is trained on the dataset with structured background, the background would be modeled. However, the keypoints can be separated into two sets by spectral clustering. (d) failure case: left) the model cannot model the occlusion well; right) the model has left and right ambiguity.

## 5   Limitations and Future Work

If the background is highly structured, the keypoints will appear on the background. Yet, we showed an avenue for future work, as already a simple graph clustering could separate the object from the background on the Human3.6M dataset. Similar to the previous 2D self-supervised methods [66, 97, 35], our model cannot model occlusion well. We show in Figure 7d left that the occluded right arm becomes the back when the person turns to the left. In addition, as for all other methods, the model cannot distinguish the left and right sides of the objects as shown in Figure 7d middle and right. We believe it is necessary to model the structure in 3D to solve these problems.

## 6   Conclusion

We presented a simple approach for learning a spatial graph representation from unlabelled image collections by reconstructing masked images. The crucial part is our learnable graph design that models the relationship between different keypoints. It is simpler than existing alternatives and opens up a path for image understanding, image editing, and learning 3D models from 2D images.

## Acknowledgement

This work was supported by the Compute Canada GPU servers, and a Huawei-UBC Joint Lab project.

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
