# OpenReview forum: "AutoLink: Self-supervised Learning of Human Skeletons and Object Outlines by Linking Keypoints"
_NeurIPS.cc/2022/Conference — NeurIPS 2022 Accept_

### Official Review · Reviewer_aWFu · 2022-07-10

**Rating:** 6
**Confidence:** 3
**Soundness:** 3 good
**Presentation:** 3 good
**Contribution:** 3 good

**Summary:**

This paper proposes to learn 2D keypoints from images, where the keypoints are linked by straight edges, from images of the same object class. The method is trained by reconstructing the original input image using masked input images and the learned edge map. The edge weights are learned and are shared across all instances in the dataset. Quantitative evaluations focus on datasets with human poses such as Human3.6M and CelebA, and qualitative results are on datasets such as Zebra, Horses, and Flowers. Results show that the proposed edge modeling is important for learning keypoints across datasets.

**Questions:**

I listed some questions above in weaknesses.

Some more questions:
- Could the authors further clarify "structured background"? Is it when the dataset have all the same background, or is it when the background have straight / patterned edges?

- What is the effect of using the masked image for appearance vs. extracting an appearance feature from the image as in [38]?

**Limitations:**

Yes the authors included limitations and potential negative societal impacts.

**Strengths And Weaknesses:**

Strengths
- To the best of my knowledge, the proposed method is new for learning keypoints in a self-supervised way with learned edge weights across keypoints.

- The evaluation is on a range of datasets, with fairly extensive ablation studies of the method with a range of hyperparameters.

- The presentation of the paper is clear, with lots of visualizations of the method, including discussions of failure cases and limitations.

Weaknesses
- The paper mentions that the training required a collection of images of the same object class (+ the graph edge weights are shared). Could the authors further discuss this requirement (perhaps in limitations in the main paper, if the class requirement is strict)? How similar do the objects need to be - would this method work on image datasets like the CUB dataset?

- The keypoints visually seem to cover object edges a lot of the time - I'm not sure if the keypoints are covering meaningful locations of the image (ex: in the provided supplementary videos & especially for flowers in Figure 6 of the appendix). Maybe this is intentional by the authors - could this be clarified? Could the authors also discuss relation of this work to works on unsupervised foreground segmentation, since it looks like the edges are covering a lot of the foreground (ex: [A])?

- The authors mention that their approach is simpler than alternatives (lines 13, 265) - however, there are simpler approaches for keypoint discovery that perform comparably on Human3.6M (ex: [39] [B] - both have fewer components than this model and doesn't need edges - they both need video though). Does the author mean that their method is simpler for image-based keypoint discovery? Also I think that Table 2 could be updated with [B] which is a self-supervised method that achieves 2.53 $\pm$ 0.056 on Human3.6M.

[A] Savarese et al., Information-Theoretic Segmentation by Inpainting Error Maximization, CVPR 2021

[B] Sun et al., Self-Supervised Keypoint Discovery in Behavioral Videos, CVPR 2022

---

> ### Author Response · Authors · 2022-08-02
> **Official comment for reviewer aWFu - Part I**
>
> ### How similar do the objects need to be?
> Objects only need to have the same topology. For example, humans have different appearances, different clothes, and different poses, but they have the same structure of a head, two arms, and two legs linked to the torso. To evaluate the limits, we already tested on AFHQ with animal faces of different species (incl. various dog breeds, domestic cats, lions, wolves, and foxes) trained together which differ significantly in texture, but they all have two eyes, two ears, a nose, and a mouth. Our model is able to learn this shared face structure for all animal faces. Furthermore, we added the experiments on CUB which includes various bird species (see Reviewer zetA).
>
> ### Keypoints focus on the boundary, intentional?
> Our initial goal was to learn the sparse skeleton structure for humans, for which this approach indeed improves results significantly (see Figure 3 and 6, by using 10 and 16 keypoints and quantitative eval.). Only when using a large number of keypoints, they focus on the boundary and, as the model is quick to train, the desired number of keypoints can be selected easily for the task at hand.  It was unintended yet we were happy with how well AutoLink scales to a large set of keypoints, still settling to semantically meaningful and consistent locations on the boundary of parts (e.g., [33] does not scale, see discussion above with reviewer zetA 2.). As our evaluation shows, sparse joint locations can also be recovered as a linear combination of the detailed results when desired, validating the semantic consistency.
>
> ### Not sure if keypoints are meaningful for Flowers?
> Flowers are a boundary case. Their shape is highly symmetric, leading to unavoidable ambiguities, similar to the left-right ambiguity discussed earlier. Moreover, different flowers differ in the number of leaves, violating our assumption on a shared topology. Therefore, positioning keypoints consistently for the same flower type on the foreground can be seen as a success.
>
> ### Discussion on unsupervised foreground segmentation methods?
> We added the discussion of the unsupervised foreground segmentation, including [A], to the related work section. It is indeed an interesting direction to extend our model to segmentation. However, more work would be needed to compete with existing segmentation methods, including finding the pixel aligned object boundaries from our relatively rough linear segments and to scale to high number of edges (currently the edge matrix has quadratic complexity and our model is limited to 1024 edges formed between 32 keypoints on a single V100 GPU.
>
> ### Aren't [39] and [B] simpler than AutoLink?
> We argue that our method is simpler in terms of network architectures, training, and application to new domains:
> [39] does not only need videos to train but also weak supervision from unpaired ground truth skeleton examples. It also requires pre-training on the skeleton-to-keypoint generation network in addition to the image generator. Besides, it uses a discriminator with CycleGAN losses, which itself can be unstable. Ours has no GAN, a single loss, trained end-to-end, and uses a single encoder/decoder pair.
> [B] does not need edges but they 1) model each keypoint as an anisotropic Gaussian distribution, with the covariance implicitly modeling part orientation and length, 2) carefully chosen different sampling rate for different videos (see their supplemental), 3) use three different losses, where two of them are later added after the keypoints are stable during training, 4) use two parallel branches to process video frames.
> By contrast, our method only has a single branch (note that the lower branch in our overview figure represents only the masking operation that is without learnable parameters and implemented in two lines of code) edges between keypoints are as easy to draw as anisotropic Gaussians, and training is on a single perceptual loss without the need to schedule the influence of multiple losses across training iterations.
> We added [B] to our table. Thank you for the pointer.
>
> ### What is a “structured background”
> With "structure" we follow a similar definition as [115], describing it as "stable local semantics across different object instances". For our object representation it means that it can be represented with the same edge weights, i.e, the topology is shared across images. Hence, a background that has common features across images that are distinctive would be considered structured and modeled as part of the edge map. Note that our spectral clustering extension still enables separation of foreground and background even when both of these properties are fulfilled. For instance, the H3.6M dataset is captured in the same room (common features) with high contrast between floor and walls (distinctive).

---

> > ### Author Response · Authors · 2022-08-02
> > **Official comment for reviewer aWFu - Part II**
> >
> > ### Using the masked image for appearance vs. extracting an appearance feature?
> > In the previous methods extracting appearance (e.g., [38]), these appearance features are not naturally disentangled from the pose (a feature encoding could store pose information). We believe this is the reason why they require manual geometric deformation or different frames in a video to disentangle. In our case, masking ~80% of pixels images removes the pose information very reliably. This reconstruction objective can be seen as a form of inpainting. As discussed in the related work section [6, 40, 51, 68, 101] indeed show that masking large regions removes structure such as pose. In addition, masking the image is a much simpler operation and therefore has a runtime advantage.

---

> > > ### Comment · Reviewer_aWFu · 2022-08-06
> > > **Thanks for the Response**
> > >
> > > I appreciate the authors for their response which addresses most of my concerns - I've updated the rating accordingly.

---

### Official Review · Reviewer_zetA · 2022-07-10

**Rating:** 6
**Confidence:** 4
**Soundness:** 3 good
**Presentation:** 3 good
**Contribution:** 3 good

**Summary:**

This work proposes a new self-supervised landmark estimation technique from image collections without any GT annotations. In contrast the existing works the discovers spatially independent landmarks, this work proposes to reason about connections/edges between the landmarks (skeletons). The core of the technique is auto-encoder with the estimated edge maps and masked images as bottleneck layers forcing the network to disentangle structure information into the learned keypoints and edges. Experiments on humans and face datasets demonstrate better results compared to existing landmark and part discovery works. Visual results demonstrate generalization of the technique to other object types (although the quality of results is not clear from the visuals on these datasets).

**Questions:**

Experiments to further strengthen the paper:
- Did authors try using unsupervised saliency to make the network focus more on foreground regions? Does the improve the results?
- If we assume a generic object skeleton is given (i.e., edge weights are given), can the proposed approach learn to attach the given skeleton to the right locations in the images?

Suggestions:
- It is better to give an intuitive explanation of edge computation (eq.3) so that authors need to look into [66] to understand this part o the paper.
- Font sizes is too small in figures-4 and 5. Increase it for better readability.

Minor corrections:
- Eq.1: Missing opening bracket in the denominator.
- Eq.3: Missing 'equal' sign.
- line 264: 'a masked images' -> 'masked images'.

**Limitations:**

Several limitations are discussed in the paper. It is better to also add a discussion on how much camera viewpoint variations does the proposed approach can handle across the image collection.

**Strengths And Weaknesses:**

Strengths:
- A simple yet effective technique for landmark discovery.
- The use of simple masking strategy to bottleneck the structure information in an image instead of learning a separate appearance encoder.
- The use differentiable edge estimation to represent structure with keypoint edges instead of just keypoints.
- State-of-the-art results on face and human datasets.

Weaknesses:
- Much of the results are shown on 2D-like object images with relatively constrained viewpoint variations. There are no quantitative results on more 3D like object datasets such as Pascal-VOC where there are 3D objects such as cars, animals from different viewpoints. For instance, existing works (e.g. [33]) report results on pascal-voc parts dataset. Does the proposed approach fail on such datasets with more 3D viewpoint variations? Paper claims the technique works well on all types of objects, but metrics are limited to only certain type of 2D objects.
- Quantitative metrics are only shown on humans and face datasets. It is difficult to assess the quality of the discovered skeletons on other object datasets (such as zebras, hands) just from visuals. Also, the visuals in Figure-1 only shows horses and zebras from very similar viewpoints.

---

> ### Author Response · Authors · 2022-08-02
> **Official comment for reviewer zetA**
>
> ### Does the model only work on 2D objects?
> Human3.6m and Taichi have highly articulated and large 3D view variations with respect to the 3D human object. Our model outperforms all existing unsupervised single-frame 2D pose estimation methods on these datasets. As we explained with examples in our limitation section, the left-right and front-back ambiguity is also a problem for all previous unsupervised 2D keypoint detectors, even when trained on videos [87], templates [85], or unpaired ground truth [39]. We argue that our method applies to 3D objects but learns to model only the visible side and in this regard has the same limitations as existing methods. Deriving a self-supervised 3D detector that models occlusions could be attempted as a post-process (e.g., using recently developed unsupervised 2D-3D lifting approaches [A]) or by extending this model to 3D. Neither direction is straightforward, e.g., [A] works merely on GT 2D poses and struggles when applied to estimated 2D poses, even when the 2D pose estimator was trained in a fully supervised way.
>
> [A] Wandt, B., Little, J.J. and Rhodin, H., 2022. ElePose: Unsupervised 3D Human Pose Estimation by Predicting Camera Elevation and Learning Normalizing Flows on 2D Poses. In Proceedings of the IEEE/CVF Conference on Computer Vision and Pattern Recognition (pp. 6635-6645).
>
>
> ### Additional comparison, e.g., to [33] on Pascal-VOC.
> A direct comparison on Pascal-VOC is not possible as their evaluation is only on foreground segmentation, not part localization. The added comparison on CUB (Table 3 revised paper, and discussion below) shows that the co-segmentation in [33] solves some of the left-right ambiguity resulting in higher scores on the seabirds where left-right matters. However, our method scores higher on the full CUB dataset, H3.6M, and CelebA  (see Table 1, 2, 3).  Furthermore, note that the sweet spots differ for both methods. On CUB they chose to use manually annotated silhouette maps to weakly supervise and their performance does not scale well to more than eight parts (as they mentioned in their supplemental) while ours remains fully self-supervised and scales to a large number of keypoints (we tried up to 32) yielding finer details.
>
> ### Difficult to assess the model with quantitative results limited to face and human?
> Faces and humans are the most commonly used domains for quantitatively testing in unsupervised 2D keypoint detection methods [30, 31, 38, 39, 59, 93, 97, 106, 85, 115]. We tested humans in different conditions (e.g., outdoor tai-chi and indoor humans on H3.6M, with and without segmented background), and included animal portraits additionally. Only a few methods [59,31,115] evaluate on CUB birds and merely on a curated version where images are mirrored so the birds point in the same direction. We added an experiment on CUB birds (new Table 3) with two protocols, one aligned and one unaligned. In the simple case when direction is aligned, all methods perform well and achieve comparable results. In the unaligned case our model outperforms all the unsupervised methods and is comparable with those using GT foreground masks [33].
>
> ### Can saliency maps help?
> We believe it is a valid idea that could possibly improve the results (see discussion of [33] above) but it is opposite to our current focus on unsupervised learning without any annotations (e.g., [33] uses GT segmentation masks as saliency maps on CUB, see their official Github [https://github.com/NVlabs/SCOPS]). We would like to explore it in our future work.
>
> ### Can the model fit a pre-defined skeleton template?
> Further constraints would be required, such as [85] using temporal data and various losses, e.g., anchor-point loss and boundary loss, and [39] requiring hundreds of example poses in a more complex cycleGAN framework. Please note that our focus is opposed to this, on learning the structure and succeeding without manually created models. Nevertheless, it is an interesting avenue for future work to integrate such a template without additional restrictions. We briefly tried to fix the connectivity in our model, but convergence to the expected body part assignments was unreliable and depended on the initialization. We believe a soft constraint towards a graph that is isomorphic to the desired template graph could succeed, but quantifying isomorphism is difficult in itself.
>
> ### Corrections and suggestions
> We added the intuition in Section 3.1 of how the edges are drawn (Gaussians extended along the edge). We enlarged the font size in Figure 4 and 5 for better readability. We corrected the missing opening bracket in Eq1, the missing 'equal' sign in Eq3, and the plural typo in line 264. Thank you for the detailed read.

---

> > ### Comment · Reviewer_zetA · 2022-08-08
> > **Follow-up discussion**
> >
> > Thanks authors for the clarifications and adding CUB results. The paper updates strengthen the paper overall.
> > - For Pascal VOC evaluation, I think, one can use Pascal-part segmentation dataset?
> > - For new CUB table, it is written that [33] uses GT silhouette supervision. I just cross-checked the work of SCOPS [33] which mentions that 'unsupervised saliency' is used (not GT silhouettes). There seem to be discrepancy with their code base. I wrote to the authors of [33] to get the clarity of their setting. They mention that the reported numbers in the paper are with unsupervised saliency, but the released code uses GT masks. Please update the table accordingly. As such the proposed method results on CUB are not that strong.

---

> > > ### Author Response · Authors · 2022-08-09
> > > **New Q&A**
> > >
> > > ### For Pascal VOC evaluation one could use the Pascal-part segmentation dataset?
> > > Thank you for the valuable pointer. We had not worked with Pascal VOC before and concluded from the foreground-background evaluation in [33] that there are no part masks available. We will consider those additional part segmentation masks in the future.
> > >
> > > ### Written confirmation from authors [33] on reported numbers in the paper are with unsupervised saliency, but the released code uses GT masks.
> > > We corrected that misclassification (Table 3, marked in red) and now report both the original results from the paper (where available) and the publicly available version reported by [15] (to be able to report on the full CUB-all dataset). Thank you for inquiring on this aspect, we did not intend to misclassify their work. Please note that we already classified it as self-supervised/unsupervised in the related work. Moreover, the other two distinguishing factors still hold: 1) We perform significantly better than [33] on Tai-Chi and CelebA-wild. 2) Ours scales better to a large number of keypoints.

---

### Official Review · Reviewer_Wivm · 2022-07-11

**Rating:** 8
**Confidence:** 4
**Soundness:** 4 excellent
**Presentation:** 4 excellent
**Contribution:** 4 excellent

**Summary:**

This paper proposed a simple yet effective framework for self-supervised learning of human skeletons and object outlines. An encoder is firstly used to regress the key points in a heatmap manner. Then a shared edge weight graph is generated by the keypoint predictions. An edge map is drawn based on the keypoint locations and edge weights. To get rid of the structure information in the image, the authors also randomly masked the original image as an appearance material to recover the input from edge maps. The model is then trained by the perceptual reconstruction loss. Experiments on human skeletons and other object outline datasets show promising results (robust key points and edge predictions).

**Questions:**

As in the Weaknesses.

**Limitations:**

Yes. The method has limitations when the background has structures or the object is occluded.

**Strengths And Weaknesses:**

Strengths:
1) This paper is well-written. The idea of this paper is novel and interesting.
2) This paper successfully built a simple yet effective self-supervised learning framework.  Different from previous self-supervised landmark prediction networks which normally enforce the key points to follow the same transformation as images, this paper constructed a shared graph and individual edge weights to reconstruct the image. The differentiable weights were inspired by the previous work on drawing and sketching, and it's very interesting to introduce them into this area for skeleton predictions.
3) To avoid the graph bottleneck from modelling appearance, this paper additionally feeds the encoder with a marked input to enrich the required appearance for reconstructing images.
4) The experimental results of this paper are impressive. By using linear regression to regress predicted facial key points to ground truths, the model achieved a very promising NRMSE result. On other additional datasets, the model is also able to predict very satisfying skeletons and key points, which cater to human perceptions. In the provided videos, the predictions are also stable in continuous frames. The predicted edge weight, which is shared in a specific task, can also well model the dependency of different key points.
5) Interesting ablation study on the number of key points and edge thickness was given.

Weaknesses:

1) The training process of this paper should be described in more detail in the main paper. For example, do you use a concentration constraint as in [115] to help learn heatmaps? Otherwise, the keypoint detector seems to be slow to converge.
2) Some figures in the paper are blurry when zoomed in (especially for the skeletons), which should be improved.
3) The authors argued that texture doesn't matter as predictions for zebra are not affected by the stripes. However, this can be because the zebra is jointly trained with horses. What if the model is only trained with objects with significant textures?

---

> ### Author Response · Authors · 2022-08-02
> **Official comment for reviewer Wivm**
>
> ### Is the concentration constraint [115] used for faster convergence?
>
> The perceptual reconstruction loss is the only one we use. One of our advantages is that we do not need any such regularizer. Still, ours is very fast to converge (3h using a single V100) which we believe is as fast or faster than related methods, e.g. [87] takes 2 days using 2 TitanX.
>
> ### More detailed training process?
>
> We believe the provided training details in Section 3.3 are complete (incl. loss, optimizer, batch size, image resolution, training time, number of iterations, and learning rate). In addition, we provided the network architecture in the supplemental Figure 1. As for the two applications, pose transfer and conditional GAN, we put the details in the supplemental Section C. We will release our code including all of them upon acceptance. If there is any missing aspect, we would be happy to discuss it further and add more details.
>
> ### Some figures in the paper are blurry when zoomed in?
>
> Since we provided hundreds of examples for each dataset, images are compressed with jpg. Is the overview figure lacking the detail? We double-checked and found the others at a reasonable trade-off between document size and quality.
>
> ### Are horses and zebras trained jointly so that they share similar structures even if they have significantly different textures?
>
> Even though joint training on different animals is possible (see animal heads experiment), we trained zebras and horses separately in this experiment. The two models resulting in nearly the same positioning shows how robust the method is. Still, we will mention that this is only one example (an extreme case, we were surprised about how well it worked). We clarified this in the revised version.

---

> > ### Comment · Reviewer_Wivm · 2022-08-09
> > **Thanks for the response**
> >
> > I am happy with the response and have no further concerns about this paper.
> > I will keep my original rating.

---

### Author Response · Authors · 2022-08-02
**Official comment for all reviewers**

We thank all reviewers for their valuable time and detailed reviews. We address the open questions below, in response to each reviewer, including evaluating on an additional dataset as requested by zetA and aWFu. We also updated the paper with additional discussion on related work and all suggested improvements, including enlarging the text in Figure 4 and 5.

Due to the 9-page limitation, we temporarily moved the ablation study details to the appendix which can be brought back with the additional page available for final versions.

---

### Meta-Review · Area_Chair_7vqG · 2022-08-26

**Recommendation:** Accept
**Confidence:** Certain

**Metareview:**

Building from works on unsupervised keypoint discovery for a domain of 2D images, this work proposes to jointly learn a skeletal structure that links discovered keypoints, and further proposes a novel image masking strategy for extracting limited background information, to force the keypoints to capture maximum information about the scene. The evaluations span a variety of datasets, with quantitative numbers on human face and body pose datasets, and show improvements from the proposed approach. A novel idea, executed well, and of interest to many at NeurIPS. Congratulations to the authors, and please fix visualization issues etc. before camera-ready / next revision.


**Award:**

No

---

### Decision · Program_Chairs · 2022-09-14

Accept